# The Effect of Functionalized Multiwall Carbon Nanotubes with Fe and Mn Oxides on *Lactuca sativa* L.

**DOI:** 10.3390/plants12101959

**Published:** 2023-05-11

**Authors:** Dorina Podar, Camelia-Loredana Boza, Ildiko Lung, Maria-Loredana Soran, Otilia Culicov, Adina Stegarescu, Ocsana Opriş, Alexandra Ciorîță, Pavel Nekhoroshkov

**Affiliations:** 1Faculty of Biology and Geology, Babeș-Bolyai University, 1 Kogălniceanu St., 400084 Cluj-Napoca, Romania; dorina.podar@ubbcluj.ro (D.P.); loredana.boza@stud.ubbcluj.ro (C.-L.B.); 2National Institute for Research and Development of Isotopic and Molecular Technologies, 67-103 Donat, 400293 Cluj-Napoca, Romania; ildiko.lung@itim-cj.ro (I.L.); loredana.soran@itim-cj.ro (M.-L.S.); adina.stegarescu@itim-cj.ro (A.S.); ocsana.opris@itim-cj.ro (O.O.); alexandra.ciorita@itim-cj.ro (A.C.); 3Joint Institute for Nuclear Research, 6 Joliot-Curie, 1419890 Dubna, Russia; pavel@nf.jinr.ru; 4National Institute for Research and Development in Electrical Engineering ICPE-CA, 313 Splaiul Unirii, 030138 Bucharest, Romania

**Keywords:** lettuce, carbon nanotubes, metal oxides, nanoparticles, bioactive compounds, antioxidant capacity, elemental content

## Abstract

The aim of this work was to evaluate the effect of six nanomaterials, namely CNT-COOH, CNT-MnO_2_, CNT-Fe_3_O_4_, CNT-MnO_2_-Fe_3_O_4_, MnO_2,_ and Fe_3_O_4_ on lettuceTo determine the impact of nanomaterials on lettuce, the results obtained were compared with those for the control plant, grown in the same conditions of light, temperature, and humidity but without the addition of nanomaterial. The study found that the content of bioactive compounds and the antioxidant capacity varied in the treated plants compared to the control ones, depending on the nanomaterial. The use of CNTs functionalized with metal oxides increases the elemental concentration of lettuce leaves for the majority of the elements. On the contrary, metal oxide nanoparticles and CNT functionalized with carboxyl groups induce a decrease in the concentration of many elements. Soil amending with MnO_2_ affects the content of more than ten elements in leaves. Simultaneous application of CNT and MnO_2_ stimulates the elemental translocation of all elements from roots to leaves, but the simultaneous use of CNT and Fe_3_O_4_ leads to the most intense translocation compared to the control other than Mo.

## 1. Introduction

Nanotechnology has numerous applications in the agricultural field because nanoparticles are advantageous for growth, development, and plant protection [1]. They can be used as nanofertilizers, nanopesticides, or herbicides [2]. Different nanoparticles with unique surface and chemical properties have been produced, carbon nanotubes (CNTs) being one of the most frequently and greatly used in different productions. Also, metal oxide nanoparticles are among the widest-used manufactured nanomaterials because of their unique properties. CNTs functionalized with metal oxide nanoparticles form a new class of hybrid nanomaterials which, in addition to the unique properties of those types of nanoparticles, also have properties due to the interactions between CNTs and attached metal oxide nanoparticles [3]. Even though, with the help of nanomaterials, the industry has advanced, there is still concern about the possibility of their release into the environment and the interaction between them and living organisms, as well as their penetration into the food chain.

In studies analyzing the effect of nanoparticles on plant growth and development, evidence of both positive and negative impacts on plants has been found. The effect depends upon the plant species and the type and concentration of nanoparticles used [2]. NPs are absorbed directly or indirectly by rootless or rooted surfaces and are transported to leaves and other parts of plants through the stem, being able to accumulate in seeds [4]. Several papers are available on the uptake of different NP and their effect on the physiology and growth of plants [5,6].

MWCNT (multiwall carbon nanotubes) have, in addition to the properties of SWCNTs (single-wall carbon nanotubes), a positive effect on the processes of germination, growth and development of plants. This positive effect in plant ontogeny is achieved by increasing the efficiency of the assimilation of minerals such as Ca and Fe, as well as increasing the efficiency of water assimilation [7]. In the case of SWCNT, it has been observed that they increase the recognition sensitivity of nitrogen monoxide [8]. 

Among the metal oxide nanoparticles used in the studies to observe their effects on plants are CeO_2_ [9], ZnO [10,11], TiO_2_ [12], NiO [13], Al_2_O_3_ [14], CdO [15], CuO [16], and Fe_3_O_4_ [17,18]. The ecosystem-level impact of accumulated nanoparticles is not well understood due to the relatively low number of studies. The continued use of chemical pesticides in agriculture is not a viable solution, but the response of biosystems to increasing amounts of NPs must be studied and documented to determine the adverse effects of these treatments on ecosystem interactions [19]. CNT-COOH can accidentally become a component of agricultural soil due to its increasingly active use in various industrial fields, such as the construction materials industry for improving the quality of cement [20] or in the electronics industry for the creation of new human sensors with high performance [21]. Also, due to its antifungal properties, CNT-COOH can become an important player in the fight against fungal plant pathogens [22] or in the stimulation of germination and seedling vigor of various species [23]. On the other hand, recent studies have shown that plant exposure to CNT-COOH can lead to an increase in the mortality of adult *A. thaliana* plants compared to the control [24] and also to an increase in *Zea mays* (maize) water transpiration [25]. There is a lack of data concerning the influence of CNT-COOH on biochemical parameters in plants.

The literature is quite limited, and most articles focus on the effect of nanomaterials on seed germination [26,27,28] and on plant growth parameters [29,30] and less on the effect on bioactive compounds or mineral composition [12,16]. Thus, the aim of our study was to evaluate the impact of Fe and Mn oxides functionalized MWCNTs, as well as of the Fe and Mn oxides, on lettuce (*Lactuca sativa* L.) growth, bioactive compounds and mineral accumulation. These nanomaterials were selected due to their possible applications for water decontamination.

## 2. Results and Discussion

### 2.1. Analysis of Plant Tissues

#### 2.1.1. Characterization of Physiological Growth Parameters

Plant development proceeded normally and uniformly, regardless of the treatments applied to the soil. During the 49 days of the growth period, no visible impact of the treatments on the phenotype of the plants exposed was observed. Visible features of a deficiency, treatment-induced toxicities, or the presence of a pathogenic attack are not present. Throughout the vegetation period, the plants were supplied with distilled water three times a week, thus, the water supply to the plants was constant and did not impose impediments in the growth of biomass. Plants retain their healthy appearance until the harvest at 49 days (Appendix A).

Average fresh biomass (Figure 1a) varied from 14.75 g to 25.49 g, the lowest value being recorded for plants grown in soil treated with MnO2 NPs, while the highest was found for the plants exposed to CNT-Fe_3_O_4_. Mean fresh biomass was statistically significantly higher than that of the control plants only for plants grown in soil supplemented with CNT-Fe_3_O_4_.

However, the mean dry biomasses of the plants (Figure 1b) grown in soil supplemented with NP were statistically significantly higher than those of the control set, except for nanoparticles containing both Fe and Mn. The plants grown in soil supplemented with CNT-Fe_3_O_4_ recorded, on average, the highest dry biomass values of 2.655 g, while the lowest average value was recorded for the lot treated with CNT-Fe_3_O_4_-MnO_2_, 1.777 g. The plants subjected to MnO_2_ NPs treatment had the second lowest dry biomass value of 1.78 g. No statistically significant differences were identified between plants treated with CNT-COOH and those treated with Fe_3_O_4_ NPs: also, those exposed to CNT-MnO_2_ are not statistically significantly different from those treated with Fe_3_O_4_ NPs.

The analysis of the H_2_O content of lettuce plants (Figure 1c) showed a statistically significant increase in plants exposed to the soil enriched with CNT-Fe_3_O_4_-MnO_2_. This high-water content of plants exposed to CNT-Fe_3_O_4_-MnO_2_ explains why the plants showed the lowest dry biomass, whereas the fresh biomass was not significantly different from the control or other treatments. No significant differences were identified between the group with the highest percentage of water and those treated with CNT-COOH and Fe_3_O_4_, and these two groups, in turn, are not statistically significantly different from the group exposed to CNT-Fe_3_O_4_. The lowest percentage of water was recorded in plants exposed to CNT-MnO_2_.

#### 2.1.2. Assimilatory Pigments Determination

Chlorophyll *a* concentration in leaves did not differ statistically significantly between different treatments and control, apart from the plants exposed to functionalized carbon nanotubes. The photosynthetic activity of the plant, from the point of view of the chlorophyll *a* concentration, is not significantly influenced by NPs (Figure 2a).

Chlorophyll *b* concentration of lettuce plants did not differ statistically significantly between treatments. The lowest and the highest average concentrations of chlorophyll *b* were recorded in plants grown in soil treated with MnO_2_ and CNT-MnO_2_-Fe_3_O_4_, respectively.

The concentration of carotenoid pigments, similar to chlorophyll *a*, was the lowest in plants exposed to CNT-COOH, being statistically different from the control group. Control plants that were not subjected to an NP treatment recorded, on average, the highest concentration of carotenoid pigments. No statistically significant differences were observed between the groups treated with iron and manganese, irrespective of the form of administration: CNT-Fe_3_O_4_, CNT- MnO_2_, CNT-MnO_2_-Fe_3_O_4_, MnO_2_, and Fe_3_O_4_.

#### 2.1.3. Total Polyphenols Determination

The total content of polyphenolic compounds in lettuce leaves was determined from the equation of the calibration curve: y = 0.5887x + 0.0058 (R^2^ = 0.9990). The obtained results, expressed as mg gallic acid/g dry weight (DW) plant material, are presented in Figure 2b.

A large variation in the total content of polyphenols was observed among the treatments. In plants exposed to CNT-COOH, the total polyphenols content was significantly higher than in control plants, whereas, in plants exposed to Fe_3_O_4_, CNT-Fe_3_O_4_ and CNT-MnO_2_, the total amount of polyphenols was significantly lower than in the control plant.

Higher content of polyphenols in plants can be regarded as an asset for increasing the quality of food and, indirectly, the quality of life through the positive effects that polyphenols have on human health. Polyphenols were shown to play an important role in maintaining the intestinal microbial balance [31], in reducing oxidative stress and in protecting beta cells against glucose toxicity (in the case of type II diabetes) [32].

#### 2.1.4. Antioxidant Activity Determination

The equation of calibration curve: y = 0.2005x + 0.0121 (R^2^ = 0.9993) was used for the determination of the antioxidant capacity of the lettuce extracts. The obtained results were estimated in mM Trolox/g DW and are presented in Figure 2c.

As in the case of polyphenols, antioxidant capacity varies largely among treatments and control plants. Compared to control plants, the antioxidant capacity is significantly higher in the lettuce grown in soil treated with MnO_2_, Fe_3_O_4_, and CNT-COOH nanoparticles, while in lettuce grown in soil treated with CNT-Fe_3_O_4_ and CNT-MnO_2_-Fe_3_O_4_ nanoparticles was significantly lower.

It is well known that a balanced diet must contain foods rich in antioxidants. These antioxidants reduce oxidative stress in the human body, thus reducing damage to cells, proteins and genetic material. Antioxidants act through several mechanisms, for example, the inactivation of metal ions by creating bonds with them and the catalytic system through which reactive oxygen species are neutralized [33].

#### 2.1.5. TEM Analysis

The method used to determine the presence of Fe_3_O_4_, or MnO_2_, is a semiquantitative one, and if these elements were not detected, it does not imply that they are absent, more that they are under the limit of detection. All samples had C and O, as these are the main elements of the living world, while Cl, K, and Ca, were present in all samples, as they are specific to plants. Other elements sporadically located were P, S, Si, Na, and Mg. Na, Mg, and Si were not detected in the untreated control. However, the plants treated with CNT and Fe_3_O_4_ had the highest levels of these elements. Traces of Na and Mg were found in one plant treated with MnO_2_, and low quantities were found in the plant treated with CNT-Fe_3_O_4_-MnO_2_. The samples treated with metal nanoparticles had low levels of Fe_3_O_4_ and MnO_2_, respectively. However, considering the administered dose (0.023 g in 100 g of substrate), these levels are justified (Figure 3).

#### 2.1.6. Elemental Content in Lettuce Leaves

Following the two types of irradiation, the maximum number of elements identified in leaf samples is twenty-three. The mass of the root samples was insufficient to divide the sample into two subsamples necessary to carry out two types of irradiation. Therefore, the decision was made to carry out only the irradiation that allows the identification of the majority of elements, being in this way required to sacrifice the determination of Mg, Al, S, Cl, and Ca, but some additional elements (La, Ta, W, and U) to those detected in leaf were identified (Table 1).

The multi-elemental composition of lettuce leaves cultivated in different environments that we found in the literature is summarized in Appendix A. The existing studies to which we refer further have very different purposes, and therefore, the published data are specific to the context of that investigation.

##### Microelements

Chlorine values reported in the literature for lettuce samples are much more homogeneous, and the differences between the samples taken in the market and those grown in polluted soils are low. Our experimental values for Cl fit well, with the range being lower than those reported by Brazilian researchers for lettuce cultivated in an urban roof garden. Only the simultaneous use of Fe_3_O_4_ and MnO_2_ associated with CNTs leads to higher content of chlorine compared to the control. All other experimental lines show values close to the control. This fact does not correlate with the previous statements that Cl^−^ ions are more readily absorbed in the presence of monovalent cations than di- and trivalent cations [34].

The range of Mn content reported in the literature for lettuce leaves is very large, from X0 of μg/kg up to X00 of mg/kg, and we cannot say that the lowest values are associated with plants collected from the market and the highest with those grown in contaminated soils. Our results are about X0 mg/kg and correspond to the middle of the above-mentioned interval. Even though MnO_2_ was added to the culture soil in three experimental treatments only when CNT-MnO_2_ was used, the Mn content of the leaves increased significantly compared to the control. The use of CNT-Fe_3_O_4_ and CNT-COOH lead to a significant decrease in Mn content in lettuce leaves.

Most available literature data for Fe content in lettuce leaves corresponds to tens and hundred mg/kg, and our results fit well with them. The only additives that affect the level of Fe are MnO_2_ and CNT-COOH, both of which cause its decrease.

Kabata-Pendias claims that Mn-Fe antagonism is widely known and is observed mainly in acidic soils that contain large amounts of available Mn. In general, Fe and Mn are interrelated in their metabolic functions, and their appropriate level is necessary for healthy plant growth.

The Co content in lettuce leaves detected in our study corresponds to data reported for cultures on contaminated soils, while the content in the market varies from less than one μg/kg to thousand mg/kg, also including our data range. All experimental lines containing MnO_2_, with or without CNT or Fe_3_O_4,_ lead to a decrease in Co content in leaves. Well-known geochemical and biochemical antagonisms between Co and Mn are based on the affinity of these metals to occupy the same sites in crystalline structures and on the similarity of their metallo-organic compounds [34]. The most significant decline takes place when MnO_2_ acts individually, without the presence of CNT and Fe_3_O_4_. This can be explained by the fact that CNT and Fe_3_O_4_ bind part of Mn, reducing part of the available Mn that competes with Co. It is stated that cobalt interacts with all metals that are associated geochemically with Fe [34], and the most significant relationship has been observed between Co and Mn or Fe in the soil and between Co and Fe in the plant. In our study, the biochemical antagonism between Co–Fe is not obvious, while the Co–Mn is evident in all experimental lines with MnO_2_.

The Zn content of lettuce leaves in different reported studies varies considerably, covering three orders of magnitude, from hundred μg/kg [35] to hundred mg/kg [36], with the peak of the dispersion around the tens of mg/kg [37,38]. Our results correspond to this peak. None of the additives significantly influenced the Zn content in the leaves.

The available data on Mo content in lettuce are quite scarce in the literature [34,39]. Our control data and those obtained in CNT-Fe_3_O_4_, MnO_2,_ and Fe_3_O_4_ experimental lines correspond to the molybdenum content of plant foodstuffs. The other three experimental lines induced a significant increase in Mo content to values that correspond to lettuce grown on industrially polluted soils [34]. The highest increase was recorded for CNT-MnO_2_ and CNT-Fe_3_O_4_-MnO_2_.

##### Macroelements

The Na content in control and experimental treatments is rather closer to the values reported for lettuce grown in polluted soils than those for lettuce available on the market. However, it should be mentioned that the latter can reach similar values, as can be seen in the lettuce from Thailand [40]. The use of CNT-MnO_2_ led to a significant increase in Na content, while other additives didn’t influence the Na content.

The range of literature available on Mg content in lettuce is very narrow, with all values corresponding to the same order of magnitude. Our results are in perfect agreement with this, varying significantly compared to the control regardless of the experimental line, except for the application of MnO_2,_ which leads to a decrease in Mg content.

There are very scarce data available for S content in lettuce which varies from 1 mg/kg up to 1.26% of S in control leave samples [41,42]. Our results fall in this range and are close to those reported by Sularz et al. [43] for control lettuce leaves in a biofortification experiment with iodine and significantly decrease when CNT-Fe_3_O_4_ and Fe_3_O_4_ are added to the soil. Kabata-Pendias [34] mentions that the interaction between Fe and S seems to be erratic in that low soil S levels may depress Fe uptake, whereas a high S content may also result in low Fe availability, depending on soil environments. Our results suggest that the enrichment of the soil with Fe can lead to a decrease in the level of S in the lettuce plant.

Values recorded for K content in control and all experimental lines are similar to those reported by Freitas et al. [37] for outer leaves of market-available lettuce and Armelin et al. [35] for lettuce grown on contaminated soils. Only exposure to CNT-Fe_3_O_4_-MnO_2_ caused a significant increase in K content.

Ca content fits well with values obtained by Freitas et al. [37] analyzing outer leaves bought at Portuguese markets. The use of MnO_2_ as an agent for CNT functionalization increases the Ca content even in the presence of Fe_3_O_4_, while the individual use of MnO_2_ decreases it.

##### Other Elements

Although the physiological function of aluminum in plants is not clear, Al is a common constituent of all plants, and it is reported to occur in plants in a large range from several mg/kg [39] up to several percent [40]. It is interesting that the values reported by Pancevki for lettuce grown on contaminated soil are much lower than those published for market-available lettuce. The mean Al content obtained for control is close to those reported by Kabata-Pendias in his review [34]. The average values obtained in three experimental lines (CNT-Fe_3_O_4_, CNT-MnO_2,_ and CNT-Fe_3_O_4_-MnO_2_) do not differ much from the control. The mean value under CNT-COOH conditions is about 35% lower than the control value, but the range of values is quite large. For this reason, we are not able to conclude that the difference is significant. When CNTs were absent, the Al content significantly decreased compared to the control.

Literature reported Rb content varies within three orders of magnitude [44], between several mg/kg to hundreds of mg/kg [38]. All our data fall into the first half of this range, and we did not notice a significant increase or decrease in the Rb content after the soil treatment. Our Br content in control and experimental lines fits well with the values reported by Pacheco et al. for market-available lettuce: these values are more than ten times lower than those reported for lettuce grown on contaminated soils or in urban areas. Like Rb, the Br content was not significantly affected by the soil amending.

There is a paucity of data on Sc distribution in plants, although Shtangeeva [45] used quite old data obtained through analysis methods that do not allow high accuracy, concluding that high concentration of Sc in plants is rather uncommon. This fact was also confirmed by Kabata-Pendias [34], who suggested that the Sc content in lettuce leaves is about 7–12 μg/kg. Bukar and Onoja [46] reported values covering a very large range from 70 μg/kg up to 37.1 mg/kg. Our results concern the control and only three of six experimental lines (CNT- Fe_3_O_4_, CNT-MnO_2,_ and CNT-Fe_3_O_4_-MnO_2_) because, for the rest, scandium was not detected. The results for amended soils do not differ much from the control and are twice lower than those determined by Bukar and Onoja [46] but higher than those reported by Kabata-Pendias [34].

The majority of reported values for As vary from several to hundred μg/kg [34,39,44] (Pachecco, Pancevski, and Kabata-Pendias), but some studies reported values of 0.31–1.04 mg/kg for lettuce grown by river irrigation [47] and even 8.76–10.1 mg/kg [48] in laboratory grown lettuce with As addition in soil or water. Our results agree with those obtained by Itanna and are highly experiment dependent. Moreover, As is the only element that varies significantly in all experimental lines. Its content rises when CNT-Fe_3_O_4_, CNT-MnO_2,_ and CNT-Fe_3_O_4_-MnO_2_ are used and decreases in all other cases. While the antagonism between Mn and As [34] could explain the decrease in As content in the presence of MnO_2_, it does not explain the increase in As content in the presence of CNT-MnO_2_. Arsenate (As(V)) is the main As species in aerobic soils. It has a strong affinity for iron oxides/hydroxides in soil [49]; thus, we may assume that when Fe_3_O_4_ was applied without CNT, part of the As in soil was fixed and lost its availability for plant sorption. MnO_2_ and Fe_3_O_4_ capture on CNT.

Very few studies refer to natural Se content in lettuce. The majority of available studies deal with the biofortification of lettuce and use hydroponic cultures, which is why the comparison is quite difficult. Nevertheless, Kabata-Pendias [34] mentions a value of 57 μg/kg and Smolen et al. [50] a value of about 1 mg/kg for different varieties of control lettuce grown on a hydroponic solution with fertilizers. Our results are about four times higher than the first value but also four times lower than the second one. The use of CNT- Fe_3_O_4_ and CNT-MnO_2_ led to an increase in Se content, while the simultaneous use of Fe_3_O_4_ and MnO_2_ to functionalize the CNT did not affect the Se content.

The Sr content in our study is about twice lower than those results obtained by Sularz et al. [43] in control lettuce, while a different attempt for biofortification of lettuce with iodine was investigated and is consistent with values reported by Pancevki et al. [39]. Both Fe_3_O_4_ and CNT-MnO_2_ soil amendment led to increased content compared to the control.

The range of values reported for Sb in lettuce leaves significantly varies by the geography of the market used for sample acquisition [37,44] but practically overlaps those reported for samples grown on contaminated soils [35]. Our results also fall in this range. The only significant variation of the Sb content compared to the control appears when MnO_2_ is used.

We are quite limited in comparison of our experimental results for Cs with literature, Cs being mostly studied in lettuce in the context of radioactive contamination of certain potential agricultural areas [51,52]. These studies are not suitable for comparison. All our values are lower than the control data reported by Armelin et al. [35]. Among six experimental lines, only the use of MnO_2_ induces a significant change (decrease) of Cs content in leaves compared to the control.

The Ba values obtained for control and experimental lettuce are similar to those reported by Pancevki et al. [39] for lettuce grown on contaminated soil, McBride et al. [53] for market-available lettuce in the USA, and Sussa et al. for urban cultivated lettuce in Brazil. The Ba content significantly decreases when MnO_2_ and CNT-COOH are used but is not affected by other additives.

As well as Sb and Cs, the Sm content was affected only using MnO_2_. Unfortunately, we couldn’t find available data for Sm content in lettuce leaves, but our control and experimental values lay in the range reported by Kabata-Pendias [34] for vegetables.

The Th content of lettuce or vegetables, which is quite rare in the literature, ranges from X or X0 µg/kg to X0 mg/kg. Our values correspond to the beginning of the interval and do not significantly change when applying different additives.

### 2.2. Root-to-Leaves Transport

To assess whether there is a preferential distribution of the determined elements towards the aerial part of the plant, the TC (translocation coefficient) was calculated for each experimental line as a quantitative ratio between the concentration of the element in the leaves of the plant and the root (Table 2).

In control conditions, the preferential distribution of elements is to root except for Na, K, Zn, and Rb. The translocation coefficient values vary between 0.04 for Sm in the Mn experimental line and 7.22 for Mo in the CNT-MnO_2_ experimental line, which is 10 times higher than the control value for this element.

As Table 2 shows, all the determined elements have a translocation coefficient higher than the control for both experimental lines that simultaneously contain CNT and MnO_2_. The CNT-Fe_3_O_4_ experimental line provides a similar result, except for two elements: Zn and Mo.

The influence on the Mo content draws attention through the very high values of the translocation factor compared to the control, especially in the presence of Mn. This phenomenon deserves a more thorough investigation. The experimental lines with Fe show a translocation coefficient lower than the control value for Mo.

The transfer of Br, Rb, Sr, and Cs is stimulated by all supplements, while the translocation of K and Na is not induced only by the use of Fe_3_O_4_. Among all the additives used, only CNT-COOH does not stimulate Ba transfer.

The most impressive ten-fold increase in the translocation coefficient we notice is for Mo during the application of CNT-MnO_2_. However, it is worth noting that most of the elements whose translocation coefficient in the control line is sub-unit, as is the case with Mo, show an impressive increase in it when the plants are grown in the presence of additives. Thus, when CNT-Fe_3_O_4_ is applied, the value of the translocation coefficient increases more than 7.5 times for Sb, and for Ba, Sm, and Th, the increase varies between 3 and 4.8 times. CNT-Fe_3_O_4_ use also induces a more than two-fold increase in TC values for Sc, Fe, and As. A similar effect for Ba, Fe, Sc, Sb, Th, Cs, Sm, and Mo has the application of CNT-Fe_3_O_4_-MnO_2_ to soil. For these elements, the value of the translocation coefficient increases from 2.3 to 6.4 times.

### 2.3. Relationship between Element Content and Bioactive Compounds and Water Content in Lettuce Leaves by Treatment

As already discussed, the use of different agents induces different variations in the elemental content, and it is also interesting to see how this correlates with the variation in the content of biological compounds and water.

The significant increase in the content of As, Se, and Ba, simultaneously with the decrease in the content of S and Mn, is accompanied by the decrease in the content of DPPH and TP in the CNT-Fe_3_O_4_ experimental line.

When CNT-MnO_2_ was applied, we noticed a decrease in TP as well as in the content of water and Co, while the content of DPPH increased along with the content of Na, Ca, Mn, As, Se, and Sr.

The use of CNT-Fe_3_O_4_-MnO_2_ induces an increase in the content of water, Cl, K, Ca, As, and Mo and a decrease in DPPH and Co.

The decrease in the content of Mg, Al, Ca, Fe, Co, As, Sb, Cs, Ba, and Sm under the influence of MnO_2_ is accompanied by the increase in DPPH, as well as the decrease in the content of S, As, and Al and the increase in Sr under the influence of Fe_3_O_4_.

The use of CNT-COOH is the only one that induces a significant change in the total carotenoid content. While the content of Mn, Fe, As, Se, Ba, and total carotenoids decreases, the content of DPPH and TP increases.

### 2.4. Cluster Analysis

Figure 4 shows the Tree Diagrams obtained on standardized data of the element content, water content and bioactive compounds for each experimental line. We considered the following options for the input data matrix: only the elemental content and only the content of bioactive compounds and water, the elemental content and one each of the six additional parameters (water content, chlorophyll *a* and *b*, respectively, total carotenoid content, total polyphenol content, and antioxidant capacity) and the totality of all these parameters.

It is obvious that the six parameters used as variables in addition to the elemental content may be divided into two groups concerning their influence on the clustering structure of the experimental lines when added individually to the pull of the elements.

The presence of the total polyphenol, water, and chlorophyll b content induces a similar pattern of clustering when the control line is closer to those using CNT functionalized with carboxylic groups and surprisingly close to the line with single MnO_2_. The lines with pure Fe_3_O_4_ and CNT-Fe_3_O_4_ form a cluster, which in turn is linked to the cluster formed by the experimental lines with CNT-MnO_2_ and CNT-Fe_3_O_4_-MnO_2_. Introducing the total content of carotenoids, antioxidant capacity and chlorophyll *a* in the data matrix one by one, we obtain another structure of clusters. The CNT-MnO_2_ and CNT- Fe_3_O_4_-MnO_2_ lines are still closely connected and form a cluster with the CNT-Fe_3_O_4_ and Fe_3_O_4_ lines, but in this cluster, the control line is connected. The connection between the CNT-COOH and MnO_2_ lines is also preserved, but not in the same way as the control line: it forms a completely separate cluster.

In all cases where the elemental content is used, the CNT-Fe_3_O_4_ and Fe_3_O_4_ experimental lines are close to each other, while the exclusive use in the analysis of biological parameters and water content separates these experimental lines into different clusters. The presence of DPPH in the input parameters is reflected in the association of the control line with the two mentioned lines. It is interesting to note that the MnO_2_ line associated with the CNT-COOH line in all analyses simultaneously presented the chemical elements and at least one of the additional parameters, while when the analysis is based only on the chemical elements or only on additional parameters, these two lines lie down in different clusters. The same can be said about the association of CNT-MnO_2_ and CNT- Fe_3_O_4_-MnO_2_ lines.

In the cluster analysis, we also applied a second tactic, not illustrated in the Figure 4. Initially, we used all parameters as input data (elemental and water content, DPPH, TP, Chl *a*, Chl *b*, and CARO), and in the following iterations, one of the parameters was eliminated. In this way, we saw that the elimination of DPPH, TP and CARO does not induce changes in the way clusters are formed. The clusters remain practically identical to those in Figure 4c, with only the linkage distance changing. Instead, the elimination of the information related to the water content induces the structure of the clusters in Figure 4d,f,g. The elimination of the information related to the Chl content modifies the structure of the clusters less radically. When the information related to the Chl content is removed, the structure of the clusters is modified less. The experimental lines CNT-Fe_3_O_4_-MnO_2_ and CNT-MnO_2_ change their position among themselves, remaining connected to the cluster containing the control line, CNT-Fe_3_O_4_ and Fe_3_O_4_. Eliminating the information on the Chl b content also influences the grouping mode of CNT-MnO_2_ and CNT-Fe_3_O_4_-MnO_2_, these forming a separate cluster associated with the first cluster that remains unchanged.

We can conclude that the increase in the number of investigated parameters is not at all to be ignored, each of them being able to contribute significantly to the conclusions drawn. For example, if we limit ourselves only to the use of the elemental content, we could conclude that the experimental line closest to the control is the one with carboxylic compounds. While considering the totality of the other investigated parameters, we see that this experimental line is the farthest from the control. We can also state that the existence of CNT in the experimental line does not lead to the appearance of a cluster that unites all experimental lines with CNT, just as its absence does not lead to a grouping of experimental lines without CNT near the control.

## 3. Materials and Methods

### 3.1. Chemicals and Materials

The ethanol and acetone used for extractions were purchased from Chimopar, Romania. For extracts characterization, Folin-Ciocalteu reagent, gallic acid, sodium carbonate anhydrous, 2,2′- diphenyl-picrylhydrazyl (DPPH), and 6-hydroxy-2,5,7,8- tetramethyl chroman-2 carboxylic acid (Trolox) were employed from Sigma-Aldrich, Germany, while methanol from Chimopar, Romania.

### 3.2. Plant Growth Conditions

For the experiments, seeds of *Lactuca sativa* variety Attraction (S.C. Agrosem Impex S.R.L., Târgu-Mureş, Romania) were cultivated in a garden substrate with active humus and fertilizer for six weeks produced by AGRO CS Slovakia, a. s. Nám. Republiky 5, 98401 Lučenec, SK, being distributed by AGRO CS Romania S.R.L. Lettuce seeds were initially germinated in water on filter paper. Four days after germination, seedlings were transferred, with two per pot to ensure the growth of at least one plant in each pot, but three days after transfer to the soil, they were weeded, leaving only one plant per pot. Pots with a diameter of 9 cm and a depth of 12 cm were used, and 100 g of soil and sand mixture were added to each pot, one-fourth of the mixture being sand and, respectively, 0.0233 g of CNT-COOH, CNT-Fe_3_O_4_, CNT-MnO_2_, CNT-Fe_3_O_4_-MnO_2_, MnO_2,_ or Fe_3_O_4_. Control plants were also grown without the addition of nanoparticles. The synthesis and characterization of the nanoparticles used in this study were reported in the previous papers [54,55,56]. The plants were grown in the phytotron, at a humidity of 60%, with a day-night cycle of 16 h light and 8 h dark, at a temperature of 20 °C/18 °C day/night. The plants were harvested seven weeks after sowing and prepared for further analysis. For each experimental variant, three individual replicates were used.

### 3.3. Determining the Effect of the Investigated Nanoparticles on the Plant Tissue

#### 3.3.1. Determination of Physiological Growth Parameters

Fresh and dry biomass of control and treated plants was determined 40 days after cultivation. To determine the dry biomass, the freshly harvested material was dried at a temperature of 60 °C for 72 h.

#### 3.3.2. Extraction and Characterization of Assimilating Pigments

The method used for the calculation of chlorophyll a and chlorophyll b, as well as carotenoid pigments, was carried out according to the protocol described by Lichtenthaler in 1987 [57]. During the first 24 h, the mass of the plant material was weighed and noted. It was then brought to a temperature of −85 °C using liquid nitrogen so that it could then be ground very quickly. The plant material was kept in the freezer at −80 °C until a 100% volume of acetone was added to the frozen material. The obtained solutions were vortexed for 10 s to ensure their homogenization. The solutions were stored in the dark for 24 h at 4 °C. On the second day, the solutions were again vortexed for 10 s and then centrifuged, the supernatant being separated. 80% acetone solution was added over the remaining plant material. The solutions were vortexed again for homogenization and stored for 24 h at 4 °C. After 24 h, the solutions were centrifuged again, and the supernatant was added on top of the supernatant collected the previous day. The quantitative determination of the assimilatory pigments was done spectrophotometrically: at 663.2 nm for chlorophyll *a*, at 646.8 nm for chlorophyll *b*, and at 470 nm for carotenoid pigments, using acetone as a reference [58].

#### 3.3.3. Preparation and Characterization of Alcoholic Extracts

The alcoholic extracts were obtained from 60% ethanol and milled dried lettuce leaves in a ratio of 1:40 by sonication for 30 min at room temperature. The obtained mixture was centrifugated for 10 min at 7000 rpm to separate the supernatant. The total polyphenol content was determined by the Folin–Ciocalteu method [57], which resulted in a blue-colored compound with a maximum absorption band at 765 nm. Consequently, 5 mL of double distilled water, 1 mL of extract and 0.5 mL of Folin–Ciocalteu reagent were added to a 10 mL graduated flask. The mixture was mixed and left to stand for 3 min, after which 1.5 mL of Na_2_CO_3_ (5 g L^−1^) was added and completed until 10 mL with double distilled water. The mixture was kept at 50 °C (in a water bath) for 16 min, after which it was cooled to room temperature. After cooling, the absorbance of the mixture was read relative to the control sample (double distilled water) at a wavelength of 765 nm, using a UV-VIS T80 spectrophotometer (PG Instruments Limited).

The total polyphenol concentration of the samples was calculated using a standard gallic acid curve for the range of 0.002–0.8 mg mL^−1^, with solutions obtained by successive dilutions from a standard solution of 1 mg mL^−1^.

The antioxidant capacity was assessed by a slightly modified procedure by Brand-Williams and collaborators [59]. Therefore, 0.01 mL of extract was added to 3.9 mL of DPPH radical solution (0.0025 g/100 mL methanol). The mixture was left in the dark for 10 min, after which its absorbance was read at 515 nm compared to the blank sample (0.01 mL extract added to 3.9 mL methanol). The results were calculated from the calibration curve that was plotted for different concentrations of Trolox (0.004–3.2 mM) and expressed in mM Trolox/g plant.

#### 3.3.4. TEM Analysis of Wheat Tissue

The dry mass of treated and control plants was ground with a mortar and pestle and pressed into a pellet. The pellet was examined with SEM Hitachi SU8230 coupled with an elemental X-ray diffraction detector (EDX). Each sample was measured in six distinct points on the pellet, and the mean was calculated.

#### 3.3.5. Elemental Content

The elemental content of the lettuce biomass (leaves and roots) was determined by neutron activation analysis (NAA) at the pulsed fast reactor IBR-2 (FLNP JINR, Dubna). A total number of 42 plant samples were analyzed. To determine short-lived isotopes, biological (about 0.3 g) were irradiated for 3 min under a thermal neutron fluency rate of approximately 1.6 × 10^13^ n cm^−2^ s^−1^. The gamma spectra were measured for 15 min. In the case of long-lived isotopes, samples were irradiated for 3 days under a resonance neutron fluency rate of approximately 3.31 × 10^12^ n cm^−2^ s^−1^, repacked and measured using high purity germanium detectors twice (after 4–5 days and 20–23 days of decay).

#### 3.3.6. Data Analysis

The calculations were done using Microsoft Office Excel 2010 (Microsoft, Redmond, WA, USA), and the graphs were performed using Origin 8 (OriginLab Corporation, Northampton, MA, USA). The values are presented as means of three parallel experiments ± standard error. The differences between means were tested for significance (p < 0.05) using one-way analysis of variance (ANOVA) followed by Tukey’s test using Minitab 17 software (Minitab Ltd., Coventry, UK).

## 4. Conclusions

The present work follows the impact of the considered nanomaterials on lettuce.

The treatment that proved to be the most effective in increasing biomass, both fresh and dry, was CNT-Fe_3_O_4_, registering the highest values in these categories.

Considering all the variables, following the analysis of the impact of carbon nanotubes enriched with Fe and Mn oxides, different effects of the same treatments on the analyzed pigments, polyphenols, and antioxidant capacity were observed. The determined amounts of chlorophyll a did not vary compared to the control, except for the group treated with CNT-COOH, which recorded a value 27.5% lower than the control group. In the case of quantitative variations between chlorophyll b recorded in the control group and the groups treated with nanoparticles, the lowest values of chlorophyll b were determined in plants exposed to MnO_2_ (15.87%) and CNT-MnO_2_ (23.8%). Also, the recorded values of carotenoids present in plants treated with nanoparticles are lower compared to the control group, the lowest concentration being determined in the case of the functionalized carbon nanotubes (47.06%).

The highest decrease in the number of total polyphenols was determined by the exposure of the plant to CNT-Fe_3_O_4_ (34.16%), and the most significant increase was recorded in the plants exposed to CNT-COOH (13.36%) compared to the control plants.

The most drastic decrease in the antioxidant capacity was determined by the exposure of the plant to CNT-MnO_2_, while the significant increase in antioxidant activity was determined in plants exposed to CNT-COOH (66.61%).

Application of CNTs functionalized individually or simultaneously with Fe_3_O_4,_ and MnO_2_ affects the elemental content by increasing it for a greater number of elements than by decreasing it. The largest number of elements is affected by soil amendment with MnO_2_.

Simultaneous application of CNT and MnO_2_ stimulates the elemental translocation of all elements from root to leaf, but the simultaneous use of CNT and Fe_3_O_4_ leads to the most intense translocation compared to control besides Mo. The most impressive ten-fold increase in the translocation coefficient we noticed was for Mo during the application of CNT-MnO_2_.

The cluster analysis highlighted three pairs of experimental lines (CNT-Fe_3_O_4_ and Fe_3_O_4_; CNT-MnO_2_ and CNT-Fe_3_O_4_-MnO_2_; and MnO_2_ and CNT-COOH) whose effect on all the investigated parameters is similar, but the degree of differentiation compared to the control line depends on the bioactive compounds.

In conclusion, the effect of carbon nanotubes enriched with Fe and Mn oxides have both a positive and a negative effect on the biochemical qualities of lettuce plants.

## Figures and Tables

**Figure 1 plants-12-01959-f001:**
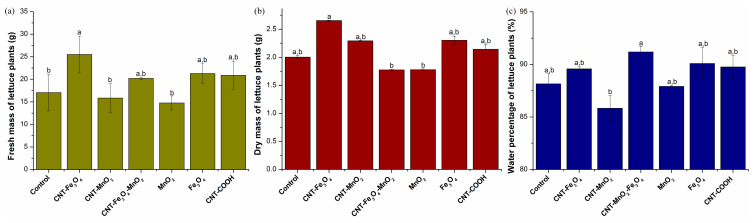
Biomass and water content of lettuce plants: fresh biomass (**a**), dry biomass (**b**) and the percentage of H_2_O in lettuce plants (**c**). Each data point is the mean ± the standard error of the mean of three independent replicates experiments; different letters mean significant differences between the treatment and the control plants, determined by Tuckey’s test (*p* < 0.05).

**Figure 2 plants-12-01959-f002:**
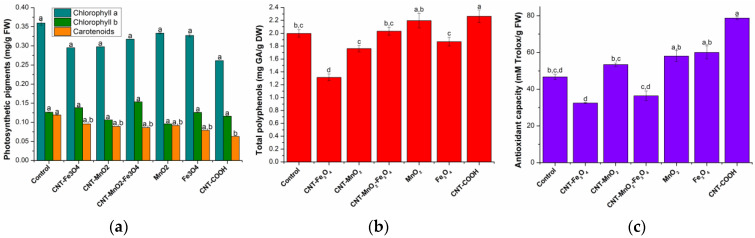
Pigment quantities (**a**), total polyphenols quantities (**b**) and antioxidant activity (**c**). Each data point is the mean ± the standard error of the mean of three independent replicates experiments; different letters mean significant differences between the treatment and the control plants, determined by Tuckey’s test (*p* < 0.05).

**Figure 3 plants-12-01959-f003:**
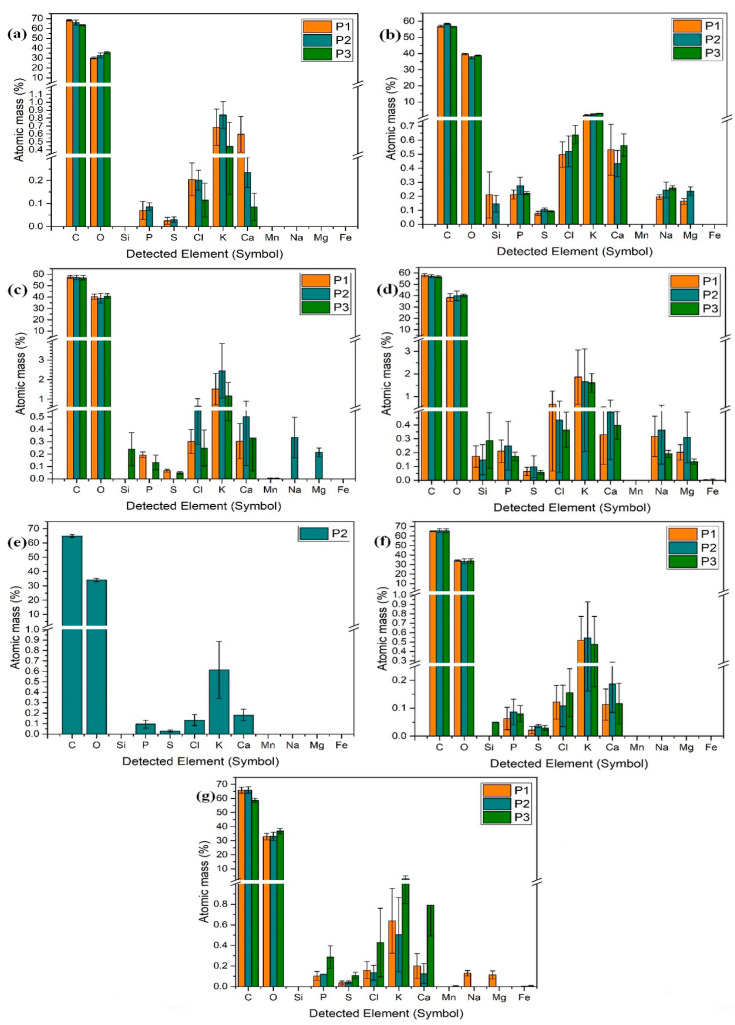
Elemental detection analysis of the plants treated with CNT-COOH (**b**), MnO_2_ (**c**), Fe_3_O_4_ (**d**), CNT-MnO_2_ (**e**), CNT-Fe_3_O_4_ (**f**), and CNT-MnO_2_-Fe_3_O_4_ (**g**), as compared to the untreated control (**a**). Except for the plants treated with CNT-MnO_2_, which had enough material only from one single replicate, all samples had three replicates (P1–3).

**Figure 4 plants-12-01959-f004:**
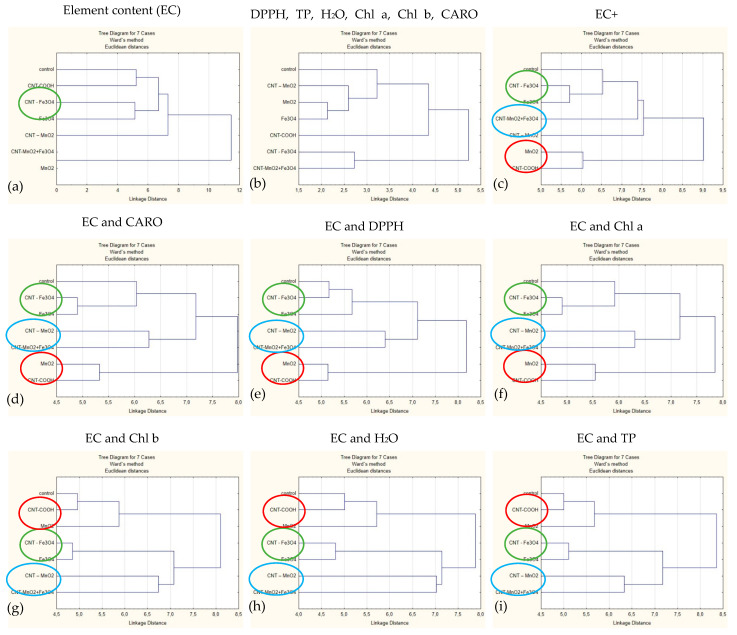
Tree Diagrams based on element content only (**a**), on water content and bioactive compounds in lettuce leaves (**b**), on all investigated parameters (**c**), on element content and each additional parameter (**d**–**i**), respectively.

**Table 1 plants-12-01959-t001:** Element content ± standard error in lettuce leaves (LL) and roots (LR) for each experimental treatment.

		Control	CNT-Fe_3_O_4_	CNT-MnO_2_	CNT-Fe_3_O_4_-MnO_2_	MnO_2_	Fe_3_O_4_	CNT-COOH
Na	LL	3.14 ± 0.34	3.83 ± 0.24	4.13 ± 0.23	3.97 ± 0.23	3.36 ± 0.31	3.82 ± 0.25	3.80 ± 0.45
	LR	1.75 ± 0.05	1.98 ± 0.25	1.90 ± 0.29	1.86 ± 0.43	1.80 ± 0.49	2.32 ± 0.24	1.80 ± 0.30
Mg *	LL	2.69 ± 0.12	2.51 ± 0.15	2.93 ± 0.18	2.86 ± 0.17	2.10 ± 0.12	2.49 ± 0.09	2.52 ± 0.14
Al **	LL	75.50 ± 16.70	75.90 ± 14.60	81.00 ± 20.60	79.70 ± 11.70	42.30 ± 3.90	45.20 ± 2.40	48.00 ± 13.50
S *	LL	5.25 ± 0.94	3.70 ± 0.58	3.93 ± 0.54	5.61 ± 1.76	3.73 ± 1.08	4.77 ± 1.24	5.53 ± 0.61
Cl *	LL	15.70 ± 1.55	13.90 ± 1.03	13.90 ± 0.54	20.10 ± 0.92	13.20 ± 2.87	15.00 ± 3.26	16.00 ± 0.80
K *	LL	89.70 ± 5.20	88.30 ± 5.67	85.70 ± 3.67	105.00 ± 3.28	90.00 ± 7.37	90.30 ± 9.87	86.00 ± 2.08
	LR	68.50 ± 5.31	60.70 ± 6.79	55.70 ± 6.98	65.00 ± 11.50	57.70 ± 8.97	71.50 ± 11.80	63.70 ± 8.69
Ca *	LL	9.15 ± 0.04	10.30 ± 0.64	12.20 ± 0.81	12.03 ± 0.32	8.27 ± 0.23	9.93 ± 0.48	9.30 ± 1.11
Sc ***	LL	35.70 ± 4.63	31.30 ± 6.17	30.30 ± 5.90	27.70 ± 3.28	-	-	-
	LR	491.00 ± 187.00	183.00 ± 50.90	207.00 ± 53.70	136.00 ± 48.40	182.00 ± 54.40	272.00 ± 148.00	217.00 ± 69.20
Mn **	LL	41.20 ± 3.10	33.60 ± 1.20	61.70 ± 8.70	43.80 ± 4.30	51.30 ± 7.80	36.80 ± 2.90	36.70 ± 0.35
Fe **	LL	190.00 ± 18.50	169.00 ± 22.30	152.00 ± 14.30	189.00 ± 29.60	60.70 ± 19.30	222.00 ± 125.00	71.00 ± 10.80
	LR	1335.00 ± 470.00	453.00 ± 84.50	740.00 ± 187.00	478.00 ± 152.00	475.00 ± 107.00	915.00 ± 388.00	673.00 ± 211.00
Co ***	LL	99.30 ± 17.40	81.70 ± 13.00	71.70 ± 7.42	79.70 ± 7.84	52.00 ± 5.57	94.00 ± 42.80	72.70 ± 15.90
	LR	680.00 ± 130.00	313.00 ± 54.30	403.00 ± 18.60	316.00 ± 61.50	396.00 ± 69.20	700.00 ± 269.00	803.00 ± 103.00
Zn **	LL	73.00 ± 11.80	60.30 ± 4.80	71.70 ± 5.80	64.70 ± 4.40	69.70 ± 5.20	59.00 ± 4.00	56.70 ± 5.50
	LR	70.00 ± 0.80	58.90 ± 20.30	46.70 ± 1.90	46.50 ± 12.60	43.80 ± 3.20	63.50 ± 7.80	49.70 ± 5.20
As **	LL	1.28 ± 0.11	2.04 ± 0.15	2.49 ± 0.36	1.49 ± 0.04	0.40 ± 0.05	0.55 ± 0.06	0.57 ± 0.07
	LR	2.38 ± 0.04	1.81 ± 0.35	1.87 ± 0.13	1.87 ± 0.41	1.68 ± 0.23	2.16 ± 0.21	1.92 ± 0.13
Se ***	LL	250.00 ± 20.00	307.00 ± 14.50	340.00 ± 11.50	223.00 ± 57.80	-	-	203.00 ± 3.33
Br **	LL	1.47 ± 0.28	1.15 ± 0.05	1.17 ± 0.04	1.24 ± 0.09	1.12 ± 0.09	1.43 ± 0.07	1.57 ± 0.19
	LR	4.70 ± 0.41	2.65 ± 0.31	3.27 ± 0.22	2.84 ± 0.74	2.70 ± 0.55	4.26 ± 1.78	4.30 ± 0.31
Rb **	LL	29.70 ± 1.45	30.30 ± 1.67	30.33 ± 1.33	36.00 ± 1.15	31.70 ± 2.73	32.70 ± 3.18	31.30 ± 1.20
	LR	25.50 ± 0.41	21.00 ± 3.06	22.30 ± 1.20	23.70 ± 3.48	21.60 ± 3.51	26.50 ± 3.70	22.30 ± 1.86
Sr **	LL	22.60 ± 1.07	25.60 ± 3.78	26.90 ± 0.98	23.20 ± 0.92	24.20 ± 2.00	26.80 ± 0.61	24.50 ± 1.50
	LR	40.50 ± 1.22	29.20 ± 2.91	30.80 ± 1.72	32.20 ± 5.14	27.40 ± 3.11	39.00 ± 4.08	30.00 ± 1.73
Mo **	LL	0.75 ± 0.19	0.46 ± 0.11	5.25 ± 2.50	4.25 ± 1.33	0.90 ± 0.27	0.59 ± 0.23	2.19 ± 0.94
	LR	1.04 ± 0.01	0.87 ± 0.08	0.73 ± 0.09	0.92 ± 0.20	0.71 ± 0.10	1.11 ± 0.16	0.93 ± 0.09
Sb ***	LL	25.70 ± 3.28	60.70 ± 32.00	24.70 ± 2.03	25.70 ± 3.84	11.90 ± 1.15	80.70 ± 63.20	16.20 ± 3.40
	LR	365.00 ± 85.70	115.00 ± 32.50	204.00 ± 31.00	126.00 ± 39.00	190.00 ± 67.10	318.00 ± 173.00	231.00 ± 63.80
Cs **	LL	20.70 ± 2.33	20.30 ± 6.89	20.00 ± 6.56	20.80 ± 3.25	12.10 ± 3.77	40.00 ± 29.60	13.70 ± 4.10
	LR	230.00 ± 82.90	62.30 ± 18.40	117.00 ± 30.50	74.00 ± 22.30	83.70 ± 23.50	132.00 ± 49.80	112.00 ± 31.80
Ba **	LL	9.87 ± 0.78	12.50 ± 1.50	11.00 ± 1.27	9.80 ± 1.71	7.10 ± 0.26	9.87 ± 2.78	6.90 ± 1.04
	LR	50.00 ± 11.40	20.30 ± 1.86	27.30 ± 4.98	21.20 ± 4.42	23.50 ± 5.09	37.50 ± 9.39	38.70 ± 6.64
La **	LR	1.54 ± 0.62	0.38 ± 0.07	0.92 ± 0.21	0.38 ± 0.12	0.44 ± 0.12	0.79 ± 0.29	0.61 ± 0.21
Sm ***	LL	18.70 ± 3.53	16.60 ± 4.13	17.50 ± 3.62	15.40 ± 2.63	2.67 ± 1.12	22.10 ± 19.40	-
	LR	260.00 ± 97.90	72.70 ± 11.85	141.00 ± 22.50	67.00 ± 26.70	71.00 ± 22.00	113.00 ± 44.10	88.30 ± 29.60
Ta ***	LR	49.50 ± 20.00	12.60 ± 3.60	22.00 ± 5.13	15.40 ± 6.97	15.60 ± 4.21	31.00 ± 18.80	24.60 ± 7.21
W **	LR	-	-	-	-	-	-	0.28 ± 0.02
Th ***	LL	21.80 ± 4.23	24.70 ± 7.88	22.7 ± 6.36	16.70 ± 0.88	-	-	-
	LR	416.00 ± 165.00	104.00 ± 17.40	239 ± 23.5	110.00 ± 38.50	129.00 ± 38.70	217.00 ± 64.90	171.00 ± 55.80

* g/kg; ** mg/kg; *** µg/kg; “-”—no available data.

**Table 2 plants-12-01959-t002:** Translocation coefficient values for each experimental treatment (TC).

	Control	CNT-Fe_3_O_4_	CNT-MnO_2_	CNT-Fe_3_O_4_- MnO_2_	MnO_2_	Fe_3_O_4_	CNT-COOH
Na	* 1.79 *	** * 1.94 * **	** * 2.17 * **	** * 2.14 * **	** * 1.86 * **	* 1.65 *	** * 2.11 * **
K	* 1.31 *	** * 1.46 * **	** * 1.54 * **	** * 1.61 * **	** * 1.56 * **	* 1.26 *	** * 1.35 * **
Sc	*0.07*	** *0.17* **	** *0.15* **	** *0.20* **	*−*	*−*	*−*
Fe	*0.14*	** *0.37* **	** *0.21* **	** *0.39* **	*0.13*	** *0.24* **	*0.11*
Co	*0.15*	** *0.26* **	** *0.18* **	** *0.25* **	*0.13*	*0.13*	*0.09*
Zn	* 1.04 *	* 1.02 *	** * 1.54 * **	** * 1.39 * **	** * 1.59 * **	*0.93*	** * 1.14 * **
As	*0.54*	** * 1.13 * **	** * 1.33 * **	** *0.80* **	*0.23*	*0.26*	*0.30*
Br	*0.31*	** *0.43* **	** *0.36* **	** *0.44* **	** *0.41* **	** *0.33* **	** *0.37* **
Rb	* 1.16 *	** * 1.44 * **	** * 1.36 * **	** * 1.52 * **	** * 1.46 * **	** * 1.23 * **	** * 1.40 * **
Sr	*0.56*	** *0.88* **	** *0.87* **	** *0.72* **	** *0.88* **	** *0.69* **	** *0.82* **
Mo	*0.72*	*0.53*	** * 7.22 * **	** * 4.63 * **	** * 1.28 * **	*0.53*	** * 2.34 * **
Sb	*0.07*	** *0.53* **	** *0.12* **	** *0.20* **	*0.06*	** *0.25* **	** *0.07* **
Cs	*0.09*	** *0.33* **	** *0.17* **	** *0.28* **	** *0.14* **	** *0.30* **	** *0.12* **
Ba	*0.20*	** *0.61* **	** *0.40* **	** *0.46* **	** *0.30* **	** *0.26* **	*0.18*
Sm	*0.07*	** *0.23* **	** *0.12* **	** *0.23* **	*0.04*	** *0.23* **	−
Th	*0.05*	** *0.24* **	** *0.09* **	** *0.15* **	−	−	−

“−” not determined; green: TC > 1; green, black: TCexp > TCcontrol.

## Data Availability

Not applicable.

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
