# Peer review of "The Effect of Functionalized Multiwall Carbon Nanotubes with Fe and Mn Oxides on *Lactuca sativa* L."

_plants, 2023, doi:10.3390/plants12101959_

Round 1

Reviewer 1 Report

The revised publication deals with an issue that has so far been little published in the professional literature. In the publication:

* the choice of nanoparticles that were used in the experiment is not sufficiently clarified - state in the introduction of the work;

* I positively evaluate the detailed description of experimental results and comparison with previously published conclusions in scientific publications;

* TEM images of nanoparticles not shown;

* describe the name of table 2 and 3 in more detail;

* the conclusion is too extensive - it is necessary to shorten it, add clear conclusions (why use modified carbon nanotubes with Fe and Mn oxide, what is the predominant positive or negative effect????, ...)

Author Response

Response to Reviewer 1 Comments

The authors would like to thank the Reviewer for his/her comments and suggestions which will be taken into account.

The revised publication deals with an issue that has so far been little published in the professional literature. In the publication:

* the choice of nanoparticles that were used in the experiment is not sufficiently clarified - state in the introduction of the work;

Response: The phrase „These nanomaterials were selected due to their possible applications for water decontamination.” Was added in introduction.

* I positively evaluate the detailed description of experimental results and comparison with previously published conclusions in scientific publications;

Response: Thank you.

* TEM images of nanoparticles not shown;

Response: The TEM images of nanoparticles are presented in the other articles. At point 3.2. Plant growth conditions, lines 523-524, is the paragraph: „The synthesis and characterization of the nanoparticles used in this study were reported in the previous papers [55-57].”

* describe the name of table 2 and 3 in more detail;

Response:

Table 2:

Initial description „Elemental content of lettuce leaves-a literature review”

New description „Elemental composition of lettuce leaves described in the literature”

Table 3:

Initial description „Translocation coefficient (TC)”

New description „Translocation coefficient values for each experimental line”

* the conclusion is too extensive - it is necessary to shorten it, add clear conclusions (why use modified carbon nanotubes with Fe and Mn oxide, what is the predominant positive or negative effect????, ...)

Response: The conclusion was shortened.

The old conclusions:” The present work follows the impact of the considered nanomaterials on lettuce.

The treatment that proved to be the most effective in increasing biomass, both fresh and dry, was CNT-Fe3O4, registering the highest values in these categories. In the case of fresh biomass, the value increased by 49.47% compared to the control, and dry biomass by 32.55%. The water concentration in plants exposed to CNT-Fe3O4 was 3.43% higher than that of the control set.

Taking into account all the variables, following the analyzes on the impact of carbon nanotubes enriched with Fe and Mn oxides, different effects of the same treatments on the analyzed pigments were observed. On the one hand, the determined amounts of chlorophyll a did not varied compared to the control, except for the group treated with CNT-COOH, which recorded a value 27.5% lower than the control group. On the other hand, in the case of quantitative variations between chlorophyll b recorded in the control group and the groups treated with nanoparticles, the lowest values of chlorophyll b were determined in plants exposed to MnO2 and CNT-MnO2. The concentrations read in these batches are respectively 15.87% and 23.8% lower than the control batch. Also, the recorded values of carotenoids present in plants treated with nanoparticles are lower compared to the control group. The lowest concentration compared to the control was determined by the activity of the functionalized carbon nanotubes, being 47.06% lower than the control.

Both in the case of total polyphenols and in the case of antioxidant activity, intensifications of the content, respectively of the activity, and decreases were observed. The highest decrease in the amount of total polyphenols was determined by the exposure of the plant to CNT-Fe3O4 (34.16%). There was no significant increase in the content of total polyphenols, the most significant increase being recorded in the plants exposed to CNT-COOH (13.36%), compared to the control plants.

The most drastic decrease in the antioxidant activity was determined by the exposure of the plant to CNT-MnO2. However, there was a significant increase in antioxidant activity in plants exposed to CNT-COOH, the antioxidant activity recorded being 66.61% higher than the control.

Application of CNTs functionalized individually or simultaneously with Fe3O4 and MnO2 affects the elemental content by increasing it for a greater number of elements than by decreasing it. On the contrary, by using MnO2, Fe3O4 and CNT-COOH, a greater number of elements suffer a decrease in content than an increase. The largest number of elements is affected by soil amendment with MnO2. There are very few regularities that we can remark concerning the relationship between element content and bioactive compounds and water content. Arsenic content variation is divergent to DPPH content change for all treatments beside CNT-MnO2. Ba and Se content, when varies, disagrees with DPPH and TP variation, respectively.

Simultaneous application of CNT and MnO2 stimulates the elemental translocation of all elements from root to leaf but simultaneous use of CNT and Fe3O lead to the most intense translocation compared to control beside Mo. The most impressive ten-fold increase in the translocation coefficient we notice of course for Mo during application of CNT-MnO2. The transfer of Br, Rb, Sr and Cs is stimulated by all additives, while the translocation of K and Na is not stimulated only by the use of Fe3O4.

The cluster analysis highlighted three pairs of experimental lines (CNT-Fe3O4 and Fe3O4; CNT-MnO2 and CNT-Fe3O4-MnO2; MnO2 and CNT-COOH) whose effect on all the investigated parameters is similar, but the degree of differentiation compared to the control line depends on the bioactive compounds.

In conclusion, the effect of carbon nanotubes enriched with Fe and Mn oxides have both a positive and a negative effect on the biochemical qualities of lettuce plants.”

became: “The present work follows the impact of the considered nanomaterials on lettuce.

The treatment that proved to be the most effective in increasing biomass, both fresh and dry, was CNT-Fe3O4, registering the highest values in these categories.

Taking into account all the variables, following the analyzes on the impact of carbon nanotubes enriched with Fe and Mn oxides, different effects of the same treatments on the analyzed pigments, polyphenols and antioxidant capacity were observed. The determined amounts of chlorophyll a did not varied compared to the control, except for the group treated with CNT-COOH, which recorded a value 27.5% lower than the control group. In the case of quantitative variations between chlorophyll b recorded in the control group and the groups treated with nanoparticles, the lowest values of chlorophyll b were determined in plants exposed to MnO2 (15.87%) and CNT-MnO2 (23.8%). Also, the recorded values of carotenoids present in plants treated with nanoparticles are lower compared to the control group, the lowest concentration being determined in the case of the functionalized carbon nanotubes (47.06%).

The highest decrease in the amount of total polyphenols was determined by the exposure of the plant to CNT-Fe3O4 (34.16%) and the most significant increase being recorded in the plants exposed to CNT-COOH (13.36%), compared to the control plants.

The most drastic decrease in the antioxidant capacity was determined by the exposure of the plant to CNT-MnO2, while the significant increase in antioxidant activity was determined in plants exposed to CNT-COOH (66.61%).

Application of CNTs functionalized individually or simultaneously with Fe3O4 and MnO2 affects the elemental content by increasing it for a greater number of elements than by decreasing it. The largest number of elements is affected by soil amendment with MnO2.

Simultaneous application of CNT and MnO2 stimulates the elemental translocation of all elements from root to leaf but simultaneous use of CNT and Fe3O4 lead to the most intense translocation compared to control beside Mo. The most impressive ten-fold increase in the translocation coefficient we notice of course for Mo during application of CNT-MnO2.

The cluster analysis highlighted three pairs of experimental lines (CNT-Fe3O4 and Fe3O4; CNT-MnO2 and CNT-Fe3O4-MnO2; MnO2 and CNT-COOH) whose effect on all the investigated parameters is similar, but the degree of differentiation compared to the control line depends on the bioactive compounds.

In conclusion, the effect of carbon nanotubes enriched with Fe and Mn oxides have both a positive and a negative effect on the biochemical qualities of lettuce plants.”

Reviewer 2 Report

The article: ‘The effect of functionalized multiwall carbon nanotubes with Fe and Mn oxides on Lactuca sativa L.’ is very interesting and deals with the current topic of high-quality food production.

The article could be shortened a bit to make the results presented more readable. The rest of the results are proposed to be transferred to the supplement. The authors described the experiment very well, but the question arises: What effect does the modification of lettuce cultivation have on living organisms?

The manuscript has sufficient scientific quality and relevance for plants. I suggest accepting the manuscript after minor revision after shortening of the article and answering the above question.

The article could be shortened a bit to make the results presented more readable. The rest of the results are proposed to be transferred to the supplement. 

Author Response

Response to Reviewer 2 Comments

The authors would like to thank the Reviewer for his/her comments and suggestions which will be taken into account.

Comments and Suggestions for Authors

The article: ‘The effect of functionalized multiwall carbon nanotubes with Fe and Mn oxides on Lactuca sativa L.’ is very interesting and deals with the current topic of high-quality food production.

The article could be shortened a bit to make the results presented more readable. The rest of the results are proposed to be transferred to the supplement. The authors described the experiment very well, but the question arises: What effect does the modification of lettuce cultivation have on living organisms?

Response: Some of the results were transferred in the „Supplementary material”. The effects of the modiffication of lettuce on living organisms will be the aim of further studies, being hard to establish only on these studies.

The manuscript has sufficient scientific quality and relevance for plants. I suggest accepting the manuscript after minor revision after shortening of the article and answering the above question.

Comments on the Quality of English Language

The article could be shortened a bit to make the results presented more readable. The rest of the results are proposed to be transferred to the supplement. 

Round 2

Reviewer 1 Report

I agree with the changes and additions to the manuscript and suggest publication.

Reviewer 2 Report

The authors introduced all proposed changes. I accept the manuscript in the present form.